# *Ace* Deficiency Induces Intestinal Inflammation in Zebrafish

**DOI:** 10.3390/ijms25115598

**Published:** 2024-05-21

**Authors:** Mingxia Wei, Qinqing Yu, Enguang Li, Yibing Zhao, Chen Sun, Hongyan Li, Zhenhui Liu, Guangdong Ji

**Affiliations:** 1College of Marine Life Sciences, Ocean University of China, Qingdao 266003, China; weimingxia@stu.ouc.edu.cn (M.W.); yuqinqing@stu.ouc.edu.cn (Q.Y.);; 2Key Laboratory of Evolution & Marine Biodiversity (Ministry of Education), Institute of Evolution & Marine Biodiversity, Ocean University of China, Qingdao 266003, China

**Keywords:** teleost, angiotensin-converting enzyme, inflammatory bowel disease, enteritis

## Abstract

Inflammatory bowel disease (IBD) is a nonspecific chronic inflammatory disease resulting from an immune disorder in the intestine that is prone to relapse and incurable. The understanding of the pathogenesis of IBD remains unclear. In this study, we found that *ace* (angiotensin-converting enzyme), expressed abundantly in the intestine, plays an important role in IBD. The deletion of *ace* in zebrafish caused intestinal inflammation with increased expression of the inflammatory marker genes interleukin 1 beta (*il1b*), matrix metallopeptidase 9 (*mmp9*), myeloid-specific peroxidase (*mpx*), leukocyte cell-derived chemotaxin-2-like (*lect2l*), and chemokine (C-X-C motif) ligand 8b (*cxcl8b*). Moreover, the secretion of mucus in the *ace^−/−^* mutants was significantly higher than that in the wild-type zebrafish, validating the phenotype of intestinal inflammation. This was further confirmed by the IBD model constructed using dextran sodium sulfate (DSS), in which the mutant zebrafish had a higher susceptibility to enteritis. Our study reveals the role of *ace* in intestinal homeostasis, providing a new target for potential therapeutic interventions.

## 1. Introduction

Chronic intestinal inflammation is a result of a breakdown in intestinal homeostasis. Intestinal inflammation occurs with many gastrointestinal diseases, among which inflammatory bowel disease (IBD) is typical. IBD includes Crohn’s disease (CD) and ulcerative colitis (UC) [1,2]. IBD occurs primarily in the mucosa of the large intestine, leading to debilitating conditions including diarrhea, rectal bleeding, and weight loss [3,4,5]. Animals with IBD have abnormal expressions of proinflammatory molecules such as IL-6, IL-1b, and TNFa and immunoregulatory cytokines such as TGFb, IL-10, and IL-35 [6,7]. Various undetermined environmental and genetic factors, even an inappropriate immune response to gut bacteria, contribute to the disease [8,9,10,11]. However, there is currently no clear pathogenesis of IBD. A widely used model to investigate the pathogenesis of IBD is the dextran sulfate sodium (DSS)-induced colitis model in mice [12,13,14,15].

Zebrafish (*Danio rerio*) have been used as animal models for human intestinal inflammation diseases including IBD [14,16,17,18]. Many genes that function in inflammatory responses are conserved between zebrafish and mammals. In mammals, for example, NOD1 and NOD2 are both involved in the detection of bacteria and contribute to gastrointestinal inflammation. In zebrafish, Nod1 and Nod2, which are expressed in intestinal epithelial cells and neutrophils, are also IBD-susceptible genes [19,20]. Other genes such as Il-1β and Sst3 also have a similar function in the innate immune response between zebrafish and mammals [21,22,23]. Therefore, zebrafish have proven to be effective models for studying intestinal damage.

The angiotensin-converting enzyme (ACE) is an essential enzyme in the renin–angiotensin–aldosterone system (RAAS), which regulates blood pressure by the cascading of enzyme proteolysis [24]. Two isoforms of mammalian ACE are recognized: the somatic form (sACE) and the germinal form (gACE) [25,26,27,28]. ACE was implicated in the pathological processes of brain ischemic injury, cardiovascular disorders, lung injury, and other processes [29]. In mice lacking *ace*, manifestations of hypotension, renal vascular thickening, and impaired urine concentration were observed [30,31,32,33]. It was reported that the apoptosis and proliferation of intestinal epithelial cells within the intestinal epithelium are compromised in mice with ACE deficiency [34,35,36]. Thus, potential correlations of ACE with inflammation in the gastrointestinal tract were demonstrated [37,38,39]. Nevertheless, the exact mechanism by which ACE exerts its regulatory effects on the gastrointestinal tract remains unclear.

Based on this, the aim of this study is (1) to understand the evolutionary conservation of zebrafish Ace among vertebrates and its specific expression in the intestine; (2) to investigate the role of Ace in the development and progression of IBD; (3) to analyze the effects of *ace* deletion in zebrafish on intestinal inflammation; and (4) to confirm the role of Ace in intestinal inflammation using a dextran sodium sulfate (DSS)-induced IBD model.

## 2. Results

### 2.1. Ace Is Evolutionarily Conserved in Vertebrates

The zebrafish *ace* gene was cloned based on the gene information from Ensembl database (Ensembl ID: ENSDARG00000079166). The Ace protein in zebrafish was identified to have two metalloproteinase domains and a transmembrane region, which are conserved among vertebrates (Figure 1A). Also, they have a similar 3-D structure (Figure 1B). The phylogenetic analysis showed that zebrafish Ace proteins are clustered with those from other vertebrates (Figure 1C), and there is syntenic conservation of *ace* between zebrafish and flameback cichlids; however, no conservation was observed among zebrafish *ace* genomic neighborhoods and those of humans and xenopus (Figure 1D), suggesting that genomic rearrangements, such as inversions or translocations, may have occurred between ray-finned fishes and tetrapods over evolutionary time, leading to the difference in gene order and synteny.

### 2.2. Zebrafish ace Expressed in Intestines during Early Development

The relative transcript levels of *ace* were examined in different tissues of adult zebrafish. We observed that the highest expression of *ace* was detected in the intestinal tissue of adult zebrafish. During the early stages of development, *ace* expression became detectable starting from 14 hpf. As the development progressed, the expression of this gene gradually increased (Figure 2C).

To further explore the spatio-temporal expression pattern of *ace* during development, WISH was performed. The results indicated that the *ace* gene was specifically expressed in the intestine of zebrafish at 4 dpf and 5 dpf (Figure 2D). This finding was corroborated by sections of the larvae following in situ hybridization (Figure 2E).

### 2.3. Ace Localized on the Cell Membrane

To investigate the subcellular localization of Ace, HEK293 T cells were transfected with either *pcDNA3.1/V5/ace/eGFP* or *pcDNA3.1/V5/eGFP* (control). Following transfection, the cells were stained with DAPI. In comparison to the control, which displayed a uniform distribution of eGFP throughout the cells, Ace exhibited distinct green fluorescence on the cell membrane and endoplasmic reticulum (Figure 3). This observation may be attributed to the transmembrane region of Ace [40].

### 2.4. Ace Deficiency Does Not Result in Intestine Defects in Zebrafish

To investigate the function of *ace*, we generated an *ace* knockout (*ace^−/−^*) zebrafish by the CRISPR/Cas9 approach. The mutation induced a frameshift in the protein-coding region, leading to premature termination of translation (Figure 4A). As expected, mRNA levels in *ace^−/−^* mutants were significantly reduced compared to those in the wild type (Figure 4B). Surprisingly, during development, the *ace*-deficient zebrafish appeared normal, exhibiting survival to adulthood and fertility without any observable morphological or developmental abnormalities. To explore whether the absence of *ace* impacts intestinal tube development, we synthesized the intestinal marker *fabp2* to probe intestinal development. The WISH analysis did not indicate any noticeable abnormalities, suggesting that the intestinal development program is not evidently affected at 5 dpf (Figure 4B). This was further confirmed by a histological examination (HE staining) of intestinal sections (Figure 4C). Thus, our results suggest that *ace* deficiency does not significantly alter intestinal development.

### 2.5. Ace Deficiency Induces Intestinal Inflammation

To investigate the differential expressed genes (DEGs) between wild-type and *ace^−/−^* mutant larvae at 5 dpf, an RNA-seq analysis was performed. The results of the Gene Ontology (GO) analysis indicated an upregulation of endopeptidase activity in the absence of *ace* (Figure 5A). Additionally, Kyoto Encyclopedia of Genes and Genomes (KEGG) analyses revealed an enrichment of DEGs in the phagosome signaling pathway, suggesting a potential involvement in immunity (Figure 5B). Notably, the DEGs also included genes expressed in the intestinal epithelium and associated with inflammation (Figure 5C). Further, *ace* is mainly expressed in leukocytes in *fxr^−/−^* zebrafish larva intestine at 6 dpf [41], which is in line with the RNA-seq results above, indicating that *ace* plays an important role in intestinal immunity (Figure 5D).

We then assessed the disparity in inflammatory cytokine expression between wild-type and *ace^−/−^* zebrafish larvae by real-time PCR. The results clearly demonstrate that the absence of *ace* substantially augments the expression of interleukin 1 beta (*il1b*), matrix metallopeptidase 9 (*mmp9*), leukocyte cell-derived chemotaxin-2-like (*lect2l*), chemokine (C-X-C motif) ligand 8b (*cxcl8b*), and the neutrophil marker myeloid-specific peroxidase (*mpx*) (Figure 6A), as well as some other validated members of the MMP families, such as *mmp9*, *mmp13a*, *mmp14b*, and *mmp30* (Figure 6B). The heightened levels of pro-inflammatory cytokines in *ace^−/−^* zebrafish indicate the occurrence of an inflammatory response. It is known that goblet cells secrete mucus to safeguard the intestines against infection, primarily concentrated in the mid- and posterior intestine regions. The abundance of intestinal mucus was further evaluated through AB-PAS staining. Interestingly, *ace^−/−^* mutants, but not the wild type, exhibited enriched mucus in their digestive tracts at 5 dpf (Figure 6C,D), indicating that goblet cells and mucus could promote intestinal defense and homeostasis. The observation of heightened mucus secretion and elevated expression of pro-inflammatory cytokines in *ace*-deficient zebrafish suggests that the deletion of *ace* may contribute to the induction of intestinal inflammation and defense.

### 2.6. Ace-Deficient Zebrafish Are Susceptible to IBD

To determine whether *ace*-deficient zebrafish are susceptible to IBD, we first used DSS to construct an intestinal infection model. Three days after drinking DSS, the expression of pro-inflammatory factors, such as *il1b*, *lect21*, *cxcl8b*, *mmp9*, and *mpx*, was increased, indicating that the intestinal infection model was successfully generated (Figure 7A,B). When zebrafish larvae were exposed to DSS at a dose between 0.5 and 1% (*w*/*v*), the survival rate of *ace^−/−^* mutants was lower than that of the wild type (Figure 7C). By HE staining on intestinal sections, we found that the intestinal tract of the *ace^−/−^* mutant larvae was significantly smaller, with a thickened wall and a smaller diameter (Figure 7D).

Furthermore, AB-PAS staining showed more mucus in the *ace^−/−^* mutants compared to wild-type zebrafish (Figure 8A). Additionally, we compared the expression of inflammatory factors between *ace^−/−^* mutants and wild-type larvae treated with DSS. Compared to that in wild-type zebrafish, the expression of *il1b*, *mmp9*, *lect2l*, *cxcl8b*, and *mpx* was significantly upregulated in *ace*-deficient zebrafish (Figure 8B). Collectively, our data indicate that *ace*-deficient zebrafish are highly susceptible to IBD.

## 3. Discussion

Angiotensin-converting enzyme (ACE), a zinc-dependent dipeptidyl carboxypeptidase composed of two metalloproteinase domains, plays a vital role in the renin–angiotensin–aldosterone system (RAAS) and is involved in immune regulation [42,43]. In mice, *ace* deficiency has been reported to affect intestinal epithelial renewal, but its precise function in intestinal inflammation remains unexplored [34]. In this study, we characterized zebrafish *ace* and found that its deletion induced intestinal inflammation, thereby expanding our understanding of ACE’s functions.

In humans, ACE contains sACE and gACE isoforms. sACE is significantly expressed in various tissues such as the small intestine, duodenum, lungs, kidneys, choroid plexus, and placenta [44,45], while gACE is specifically expressed in testes and is associated with male fertility [46,47]. Notably, the highest expression level of sACE was observed in the small intestine [48]. This is in line with our finding in zebrafish that the expression of the *ace* gene is highest in the intestine. However, an analysis of tissue expression revealed a lack of notable expression in the testes in zebrafish, and we also found that both female and male *ace* mutants’ reproduction was not affected when compared to the wild type; this is different from the gACE detected in human, suggesting that there is no functional gACE in zebrafish.

Although the absence of *ace* appears to have no discernible impact on the zebrafish’s intestinal development as indicated by its morphology, transcripts of the *ace* gene were detected as early as 14 hpf and continued to be expressed throughout development in different adult tissues including the spleen and intestine. The zebrafish begins to form a digestive tract in a segmental fashion at the mid-somite stages (18 hpf) and completes gut tube morphogenesis at 34 hpf; after the onset of exogenous feeding (5 dpf), there is a dramatic increase in the size of the intestine and the appearance of epithelial cells and other cell types [49]. In accordance with this process, from 4 to 5 dpf, *ace* specifically localized in the intestine. Moreover, a single-cell transcriptome analysis of zebrafish intestine at 6 dpf showed that *ace* expressed in many cell types, including the highest expression in leukocytes and moderate expression in enteroendocrine cells (EECs), enterocytes, goblet cells, ionocytes, and secretory precursors [41], suggesting that *ace* may play an important role in intestinal homeostasis and immunity.

More mucus was secreted in the mid- and posterior intestine of *ace^−/−^* mutants compared to the wild type at 5 dpf, indicating the occurrence of intestinal inflammation. Further, the transcriptome sequencing of 5 dpf larvae revealed that the absence of *ace* had a significant impact on immune pathways, leading to a notable increase in the expression of proinflammatory factors including *il1b*, *lect21*, *cxcl8b*, *mpx*, and *mmp9*, as well as several other members of the *mmp* gene family, such as *mmp9*, *mmp13a*, *mmp14b*, and *mmp30*. It has been observed that some of the MMP family members are associated with various inflammatory responses [50,51]. Thus, there is a potential correlation between intestinal inflammation and the deletion of *ace*. This correlation was further supported by the constructed IBD model, where intestinal inflammation induced by DSS showed a similar expression pattern of proinflammatory factors to that observed in *ace^−/−^* mutants. Moreover, compared to the wild-type larvae, the *ace^−/−^* mutants exhibited a significantly lower survival rate, increased mucus secretion in the intestine, and a notable upregulation of various inflammatory factors in this IBD model, suggesting that both *ace* deletion and DSS induction synergistically contribute to the occurrence and progression of inflammation. Although ACE in wild-type larvae is mildly upregulated in a dextran sodium sulfate colitis model [52], it is possible that *ace* deficiency impedes the ability of larvae to mount an appropriate immune response and increases their vulnerability to DSS stimuli, ultimately resulting in earlier mortality compared to the control group.

Many studies have documented the relationship between ACE and inflammation in mammals, and some results seem controversial. For example, ACE overexpression in myeloid-derived cells has been shown to increase the production of pro-inflammatory cytokines, such as IL-12β, TNF, or nitric oxide, while ACE overexpression in neutrophils has been shown to increase resistance to infections with MRSA, Klebsiella pneumoniae, and Pseudomonas aeruginosa [40]. *Ace^−/−^* mice have a less vigorous immune response to MRSA infection [40]. However, ACE-overexpressing macrophages have been shown to attenuate neuropathology and neuroinflammation [51]. Although ACE inhibitors have been shown to reduce vascular inflammation, there is no convincing evidence indicating that ACE inhibitors reduce plasma levels of major inflammatory markers in hypertension models [53]. This controversy is partly attributed to the possibility that inflammatory responses may be controlled by local rather than global immune, vascular, and inflammatory cell responses to infection or injury [54], with other localized factors involved in the inflammation process.

In this study, ACE was specifically localized in the intestine and was highly expressed by leukocytes present in intestinal tissue. A deficiency of ACE induced a mild inflammatory response in the intestine of zebrafish, indicating a potential role for ACE in immune regulation and gut health. It appears that other factors also participate in the *ace*-deficiency-induced inflammatory response, which are likely to depend on immune cells and molecules at the site of tissue damage or infection.

## 4. Materials and Methods

### 4.1. Ethics Statement

Embryos were produced through natural mating. All zebrafish studies were conducted according to the Animal Care and Use Committee of the Ocean University of China (SD2007695).

### 4.2. Zebrafish Strains and Mutants

Zebrafish (*D. rerio*) from the AB strain were kept at 28 °C and fed twice daily during dark periods of 10 h and 14 h. The fragments of *ace* were amplified using the specific primers S1 and AS1 by PCR (Table 1). Using CRISPR/Cas9 technology, *ace^−/−^* mutant lines were derived from the AB line of zebrafish. This study used 5′-TGGCTTCCATGAGGCCATCG-3′ in exon 8 as the knockout target. Mixtures of Cas9 mRNA and targeting gRNA were microinjected into one-cell-stage zebrafish embryos. A comparison with the wild-type zebrafish sequence confirmed the mutation sites. Fish from the F1 generation carrying a 7 bp deletion were crossed to obtain the F_2_ generation [55,56].

### 4.3. Bioinformatics Analysis

A set of 12 ACE sequences was primarily collected from National Center for Biotechnology Information (NCBI) website (http://www.ncbi.nlm.nih.gov/). Protein domains were predicted by the SMART website (http://smart.embl.de/); protein 3-D structures were predicted by Alphafold (https://alphafold.com/). A phylogenetic tree was constructed using IQ-tree with the maximum-likelihood algorithm, the L+G4 model, and 1000 bootstrap replications.

### 4.4. Whole-Mount In Situ Hybridization (WISH) in Zebrafish Larvae

The zebrafish embryos for WISH were cultured from 12 h post-fertilization (hpf) in E3 embryo medium containing 0.004% PTU (Sigema, P6148, Fukushima, Japan). The fragments of *ace* were amplified using the specific primers S2 and AS2 by PCR (Table 1). The fragments were digested with Sph I (Takara Bio, 1246S, Kusatsu, Japan), and Sp6 RNA polymerase (Thermo Scientific, EP0131) (Waltham, MA, USA) was used to synthesize an antisense probe labelled with digoxigenin (DIG) (Sigema, 11277073910). WISH followed the protocol in its procedures [57]. Stereomicroscopy (Nikon, SMZ1270/1270i, Tokyo, Japan) was used to observe and photograph stained embryos.

### 4.5. RNA Extraction and RT-qPCR from Zebrafish Samples

Zebrafish total RNA was extracted using Total RNA Kit I (Omega, R6834-02) (Biel/Bienne, Switzerland). DNase was used to treat the RNA, and PrimeScript^TM^ RT reagent with gDNA Eraser (TaKaRa, RP047A) was used to synthesize the cDNA. As controls, reverse transcriptase-free samples were added. Quantitative PCR was performed on an ABI 7500 machine with ChamQ SYBR Color qPCR Master Mix (Vazyme, Q431-02) (Nanjing, China). To normalize the data, the *β-actin* gene was used as the internal reference gene. A comparative Ct method (2^−ΔΔCt^ method) was used to calculate relative expression levels. All quantitative PCR experiments were conducted in triplicate. This study used the primers shown in Table 1.

### 4.6. AB-PAS Staining and HE Staining in Zebrafish Larvae

Ice-cold acetone was used to fix zebrafish embryos. Wuhan Servicebio Technology Company performed microtomy and staining of the embryos. In each section (8 μm) after AB-PAS staining, mucus-containing areas of the staining in the mid-intestine and posterior intestine were quantified using ImageJ software (1.48v). The image processing followed the protocol in its procedures [58].

### 4.7. RNA-Seq Analysis

RNA was extracted from both wild-type and *ace^−/−^* larvae at 5 days post-fertilization (dpf). The same batch of samples was sequenced for both control and knockout groups. The transcriptome was sequenced using Novogene (Beijing, China). The gene abundance was calculated and normalized using RPKM (reads per kb per million reads). DEGs (differentially expressed genes) between groups were determined using the EdgeR package (http://www.r-project.org/). Differential gene expression analysis was conducted with a fold change criterion of ≥2 or ≤0.5, and a false discovery rate (FDR) of <0.05 was considered significant. Subsequently, the KEGG pathways (Kyoto Encyclopedia of Genes and Genomes) were analyzed for enrichment. The single-cell profile of ACE in *fxr*^−/−^ zebrafish larvae at 6 dpf was generated using the Single Cell Portal website (https://singlecell.broadinstitute.org/single_cell/study/SCP1675/zebrafish-intestinal-epithelial-cells-wt-and-fxr?genes=ace&tab=distribution#study-visualize, accessed on 1 December 2023), and the raw data can be accessed from the NCBI GEO Database: GSE173570 [41].

### 4.8. DSS Treatment

Analyses of DSS (Yeasen Biotech, 60316ES25) effects were conducted on wild-type and *ace^−/−^* larvae. All embryos (n = 26–32 larvae per concentration) were cultured to 3 dpf in E3 embryo medium before the tests. From a 10% DSS stock solution (*w*/*v*), dilutions were prepared (0.5%, 0.6%, 0.7%, 0.8%, 0.9% *w*/*v*, 1.0%) with the medium replaced daily. After 3 days of treatment, the survival rate was calculated. Later experiments (0.5% DSS) used the lowest toxicity dosage. From 3 dpf to 6 dpf, the wild-type and *ace^−/−^* DSS groups were cultured in 0.5% DSS in E3 embryo medium, while the corresponding control groups were cultured in parallel in E3 embryo medium only. The RNA was isolated from half of the larvae at 6 dpf, and the other half were used for histopathological analysis [23,59].

### 4.9. Subcellular Localization

To examine the Ace protein localization in subcellular compartments, a subcellular localization assay was conducted. The *pcDNA3.1/V5-His A* plasmid was used to create the *pcDNA3.1/V5/eGFP* vector by cloning eGFP. Next, with primers that contain EcoR I and EcoR V restriction enzyme cutting sites, the complete coding region of *ace* was amplified by PCR using specific primers (S5, AS5). Recombinant plasmids were constructed by inserting the fragments upstream of *pcDNA3.1/V5/eGFP*. To investigate the subcellular localization of Ace in HEK293 T cells, the *pcDNA3.1/V5/ace/eGFP* and *pcDNA3.1/V5/eGFP* recombinant plasmids were separately transfected. DAPI staining was performed 24 h post-transfection, and observations were made using confocal microscopes after washing the samples with PBS [60].

### 4.10. Statistical Analysis

Statistical analyses were conducted using Graphpad Prism 9.0.0 software. The data are presented as the mean ± standard deviation. Student’s *t*-test was used for statistical analysis. *p* < 0.05 was considered statistically significant.

## 5. Conclusions

In summary, we demonstrated the evolutionary conservation of zebrafish *ace* among vertebrates, with specific expression observed in the intestine. Interestingly, deficiency in *ace* does not lead to intestinal defects in zebrafish but rather triggers intestinal inflammation. Furthermore, zebrafish lacking *ace* show heightened susceptibility to inflammatory bowel disease (IBD) phenotypes. Future studies are warranted to identify the particular cell types impacted by Ace deficiency, along with the specific signaling pathways in which Ace is involved in innate immunity. This understanding holds significant potential for advancing novel therapeutic strategies for inflammatory bowel diseases in human patients.

## Figures and Tables

**Figure 1 ijms-25-05598-f001:**
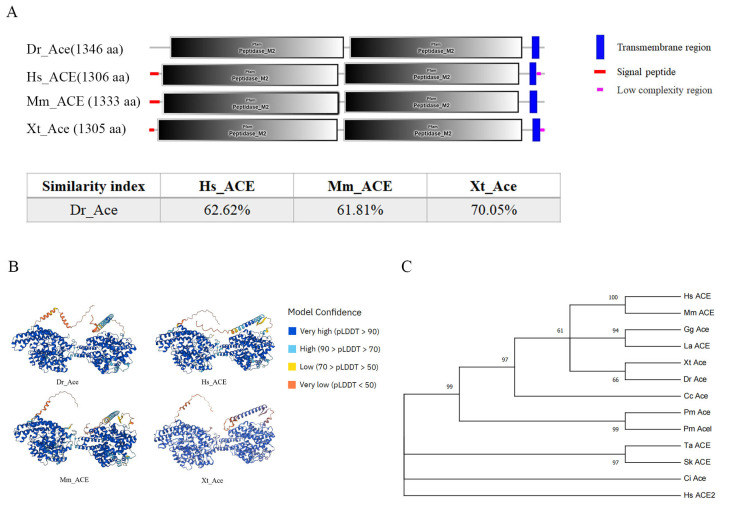
Homology comparison of Ace homologues in different species. (**A**) The secondary structures of Ace homologues in *D. rerio*, *H. sapiens*, *M. musculus*, and *X. tropicalis*. This diagram was generated using SMART online software (http://smart.embl-heidelberg.de/, accessed on 1 December 2023). Red squares, signal peptides; blue rectangles, transmembrane regions; M2, metallopeptidase 2 domain. The similarity index was derived by DNAMAN software (9.0.1.116). (**B**) The three-dimensional structures of *D. rerio* ACE, *H. sapiens* ACE, *M. musculus* ACE, and *X. tropicalis* ACE. This diagram was generated using the Alphafold online server (https://colab.research.google.com/github/deepmind/alphafold/blob/main/notebooks/AlphaFold.ipynb accessed on 5 December 2023). AlphaFold produces a per-residue model confidence score (pLDDT, predicted local distance difference test) between 0 and 100. Some regions below 50 pLDDT may be unstructured in isolation. (**C**) A phylogenetic tree of Ace homologues of various species was constructed with IQ-tree using the maximum likelihood method. Hs: *H. sapiens* (NP_000780), Mm: *M. musculus* (NP_997507), Gg: *G. gallus* (NP_001161204), La: *L. agilis* (XP_033024785), Xt: *X. tropicalis* (NP_001116882), Cc: *C. carcharias* (XP_041029546), Dr: *D. rerio* (XP_694336), Pn: *P. nyererei* (XP_005721704), Lo: *L. oculatus* (XP_015217378), Pm: *P. marinus* (XP_032821231, XP_032807619), Ci: *C. intestinalis* (XP_026693950), Sk: *S. kowalevskii* (XP_002741143), and Ta: *T. adhaerens* (XP_002111333). Each node was bootstrapped with 1000 replications to estimate its reliability. (**D**) Syntenic analysis among zebrafish *ace* genomic neighborhoods and those of humans, frogs, and flameback cichlids. Different colors indicate different genes. Orthologs of these genes in other species are shown in corresponding colors.

**Figure 2 ijms-25-05598-f002:**
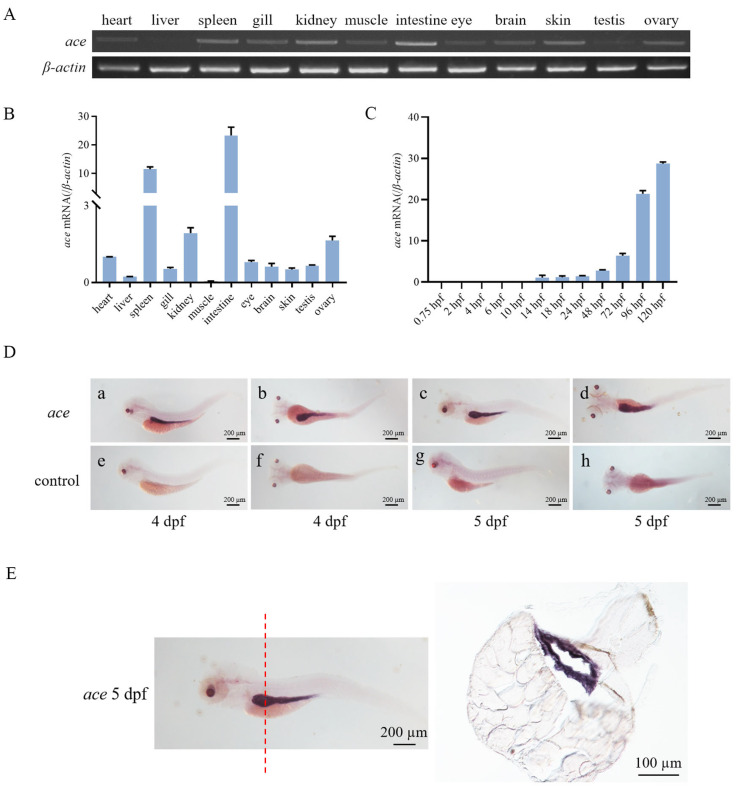
Expression patterns of *ace* mRNA in zebrafish. (**A**) Transcriptional expression patterns of *ace* mRNA in different tissues (heart, liver, spleen, gill, kidney, muscle, intestine, eye, brain, skin, testis, and ovary) were detected by RT-PCR. *β-actin* was used as an internal control. (**B**,**C**) Relative transcriptional expression of *ace* in different tissues and different developmental stages as measured by RT-qPCR. hpf: hours post-fertilization. *β-actin* was used as an internal control. (**D**) Spatio-temporal expression patterns of *ace* in embryos detected by WISH. “a–d” represents the results of anti-sense probe hybridization, while “e–h” represents the results of sense probe hybridization (negative control). dpf: days post-fertilization. (**E**) The intestinal region of the larvae at 5 dpf after WISH was frozen and sectioned (8 µm). The red line indicates the position of the frozen section.

**Figure 3 ijms-25-05598-f003:**
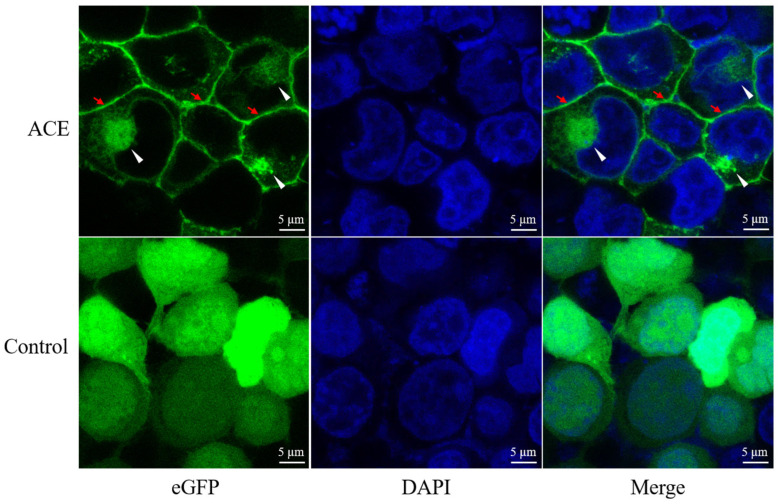
Subcellular localization of Ace in HEK293T cells. Recombinant plasmids *pcDNA3.1/V5/Ace/eGFP* and *pcDNA3.1/V5/eGFP* (control) were transiently transfected into HET293T cells. DAPI stained the nucleus. Red arrows point to the cell membrane and white arrowheads point to the endoplasmic reticulum.

**Figure 4 ijms-25-05598-f004:**
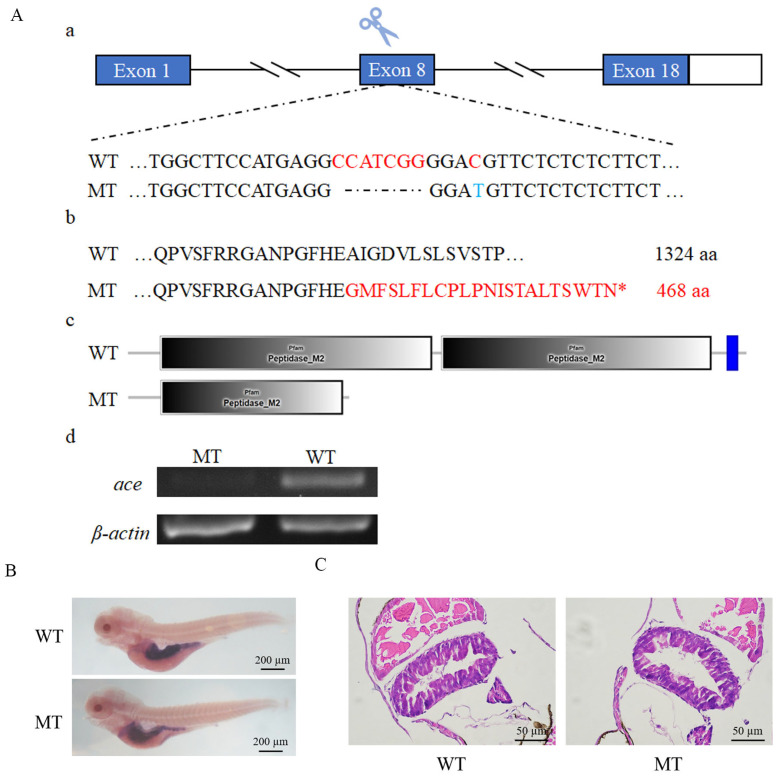
Knockout of ace and morphological examination of ace^−/−^ mutant larvae. (**A**) The generation of ace mutations in zebrafish through CRISPR/Cas9 technology. (**a**). The targeted exon 8 of ace containing the knockout site. Exons are indicated by boxes, blue box means coding region of exon, while white box means non-coding region of exon, introns are represented by fold lines. The red letter in the wild type (WT) means the position some changes will happen, while the dotted line and blue letter corresponded to red letters means deletion and transition happened in the mutant (MT), respectively. (**b**,**c**). In comparison to the wild type, the mutants had a deletion of 7 nucleotides in exon 8, leading to skipped mutation (red letters), and premature termination of translation at the 468th amino acid. The asterisk in the figure indicates the position of protein translation termination. (**d**). To assess the impact of ace knockout, RT-PCR was performed to analyze ace expression in both ace^−/−^ and wild-type samples (**B**) Expression patterns of fabp2 in ace^−/−^ mutants and wild-type larvae at 5 dpf, as detected by WISH. (**C**) The morphology of the intestinal epithelium in ace^−/−^ mutants and wild-type larvae at 5 dpf, as shown by HE staining.

**Figure 5 ijms-25-05598-f005:**
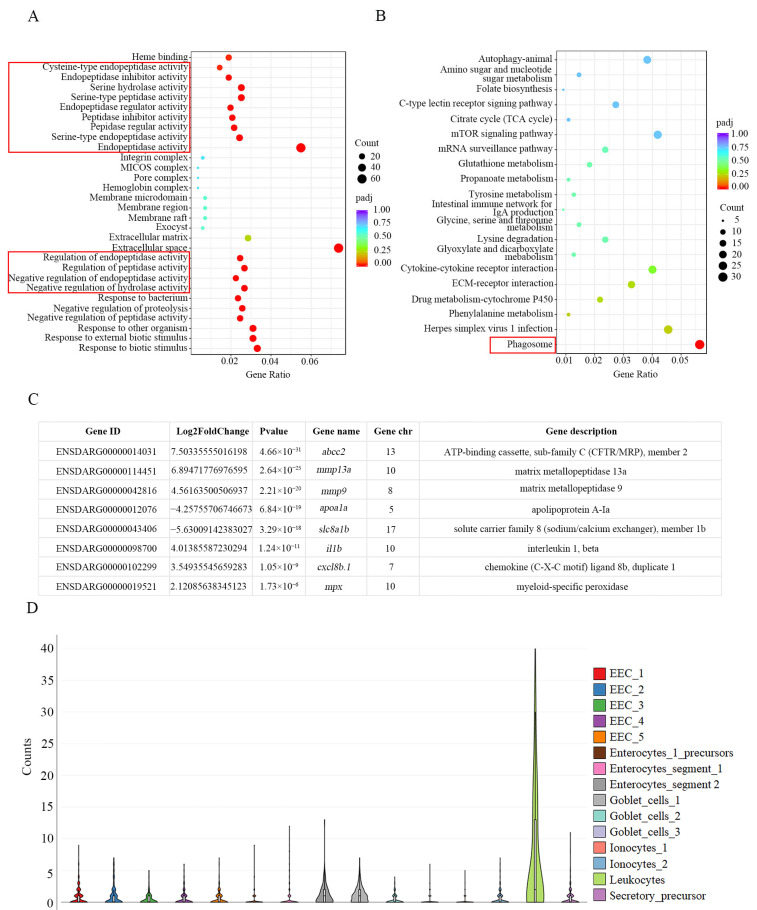
RNA-seq analysis of *ace^−/−^* mutants and wild type at 5 dpf. (**A**,**B**) GO and KEGG analysis of the pathways through the enrichment of DEGs between *ace^−/−^* mutants and wild type. The red box signifies the peptide enzyme or immune-related signaling pathways. (**C**) The table shows some DEGs with significant changes, including genes highly expressed in intestinal epithelium and some immune genes. (**D**) Single-cell profile of ACE in *fxr^−/−^* zebrafish larvae at 6 dpf.

**Figure 6 ijms-25-05598-f006:**
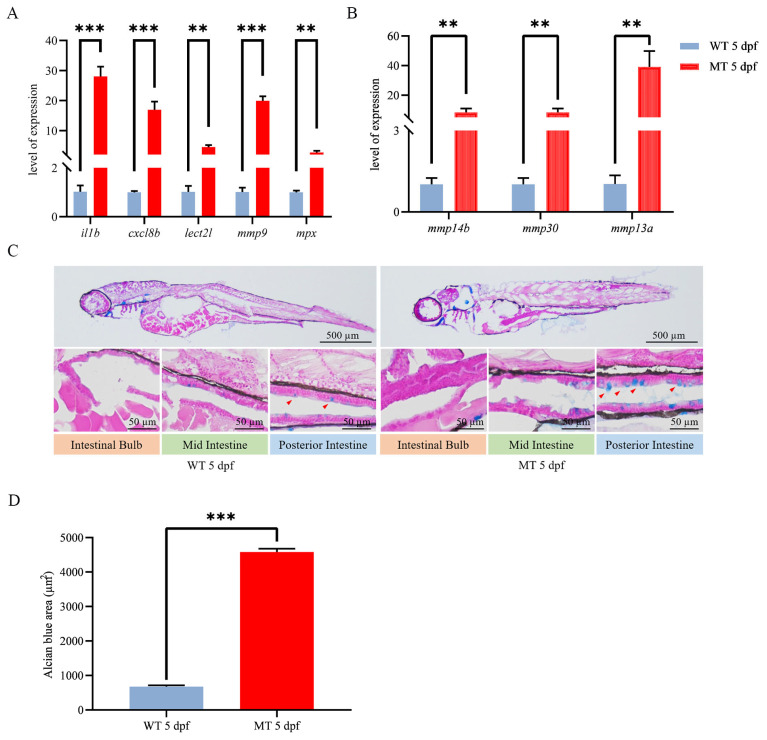
Inflammatory response evaluation for *ace^−/−^* mutants and wild-type larvae at 5 dpf. (**A**,**B**) qRT-PCR analysis of selected inflammation-related genes. (**C**) AB-PAS staining sections label mucus in the intestine of both *ace^−/−^* mutants and wild-type larvae at 5 dpf (upper panel). The lower panels provide enlarged photographs of the intestinal bulb and mid- and posterior intestine regions. The red arrowhead in the figure indicates the staining signal. (**D**) Cell statistics for Alcian blue staining in the digestive tract of *ace^−/−^* mutants and wild-type larvae at 5 dpf. Data are presented as the mean ± SD of 10 fish. ** *p* < 0.01, *** *p* < 0.001. The experiments were independently repeated three times.

**Figure 7 ijms-25-05598-f007:**
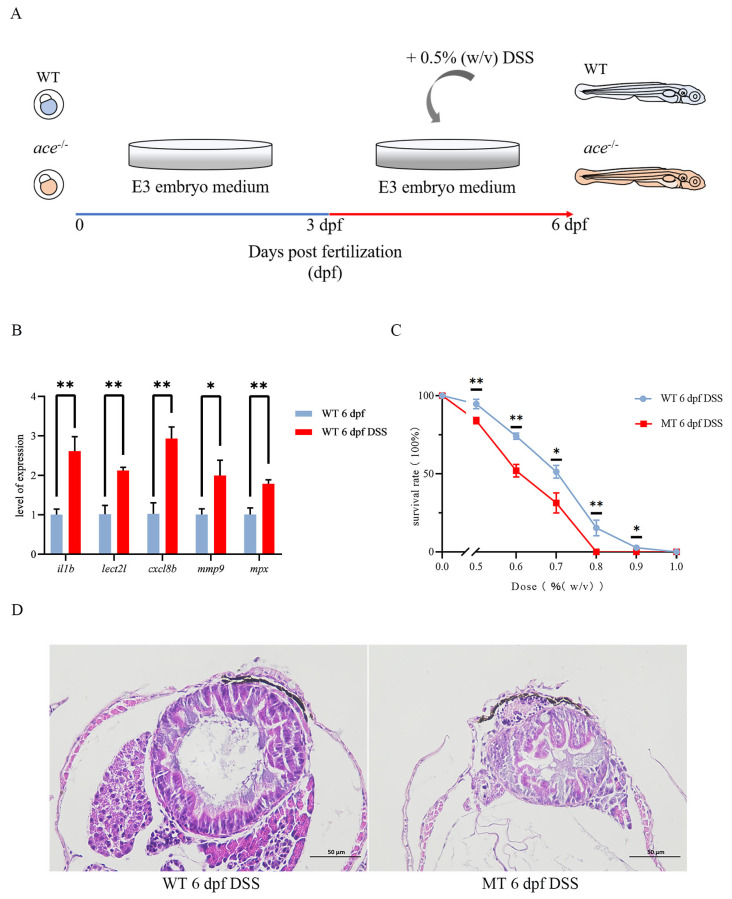
The IBD model was successfully constructed using DSS. (**A**) A diagram of IBD model development. (**B**) Changes in DSS-treated and untreated wild-type larvae were examined by qRT-PCR for several pro-inflammatory factors: *ilb*, *lect2l*, *cxcl8b*, *mmp9*, and *mpx*. Data represent the mean ± SD. * *p* < 0.05, ** *p* < 0.01. Three independent biological replicates were performed. (**C**) Line chart of survival rates comparing *ace^−/−^* mutants and wild-type larvae treated as the DSS dose (% (*w*/*v*)) increased from 0% to 1% (n = 50). The significance of differences is annotated above the nodes in the line chart. * *p* < 0.05, ** *p* < 0.01. (**D**) HE staining shows a more severely disorganized intestinal epithelium in *ace^−/−^* mutants compared with the wild type at 6 dpf following DSS treatment.

**Figure 8 ijms-25-05598-f008:**
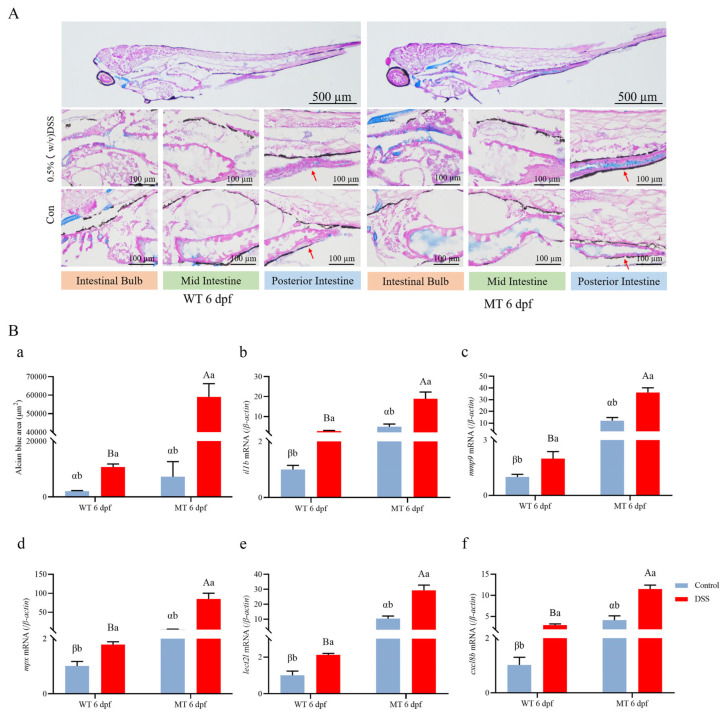
Histopathological assessment comparing *ace^−/−^* mutant and wild-type larvae treated with DSS. (**A**) AB-PAS staining sections highlighting mucus in the intestine of DSS-treated and untreated *ace^−/−^* mutants and wild-type larvae (upper panel). The lower panels show enlarged photographs of the intestinal bulb and the mid- and posterior intestine regions. At the bottom, the control group exhibits AB-PAS staining sections that did not undergo DSS treatment. The red arrow in the figure indicates the staining signal. (**B**) (**a**). Quantification of Alcian blue-stained cells in the digestive tract of *ace^−/−^* mutants and wild-type larvae at 6 dpf. Data represent the mean ± SD of 10 fish. (**b**–**f**). The expression of several pro-inflammatory factors in *ace^−/−^* mutants and wild-type larvae examined by qRT-PCR. Data are presented as the mean ± SD. Different lowercase letters indicate significant differences (*p* < 0.05) within the group; different uppercase letters indicate significant differences (*p* < 0.05) between the groups after DSS treatment; different Latin letters indicate significant differences (*p* < 0.05) between the groups before DSS treatment.

**Table 1 ijms-25-05598-t001:** Primers used in this study.

Name	Sequence (5′-3′)	Sequence Information
S1	CGCAACCAGTAACTACGCATT	For *ace* knockout (*ace*)
AS1	CTCTCCACTGAACACACTCCAT	
S2	CTCCACAGCATTGACCTCCT	For WISH (*ace*)
AS2	GTAAGCGTTCCAGACCTCCT	
S3	CGCTCTACCTCAGCGTTCAT	For RT-qPCR (*ace*)
AS3	GCCCACATGTTTCCAAGCAG	
S4	GGTATTGTGATGGACTCTGGTGAT	For RT-qPCR (*β-actin*)
AS4	TCGGCTGTGGTGGTGAAG	
S5	CCGGAATTCATGAACAGAGGGAAAAGGGAGAG	For recombinant (*ace*)
AS5	CCGGATATCCCTTGAGCTCCATCTGAGACATGG	
S6	GATCCGCTTGCAATGAGCTAC	For RT-qPCR (*il1b*)
AS6	TCAGGGCGATGATGACGTTC	
S7	TCAGGGCGATGATGACGTTC	For RT-qPCR (*mmp9*)
AS7	TAGCGGGTTTGAATGGCTGG	
S8	CCAGAACCAGTGAGCCTGAG	For RT-qPCR (*mpx*)
AS8	ACTCTCTTCTTCTGCCCCCA	
S9	TAGCTTGAGTGGAGGAGGTCT	For RT-qPCR (*lect2l*)
AS9	CATGGGAAGTGATGCCAGGA	
S10	GTGCGCCAATGAGGGTGAA	For RT-qPCR (*cxcl8b*)
AS10	ACCCACGTCTCGGTAGGATT	
S11	CGCCAACAACCAGGTTTACAGTTAT	For RT-qPCR (*mmp13a*)
AS11	TCAGGACGCGTAACAGCTTG	
S12	ACAATGCGTTCTGTCGATGC	For RT-qPCR (*mmp14b*)
AS12	AGCGGGAGAATACAGACGTT	
S13	GCAGCTCTCATCCTTGTGGT	For RT-qPCR (*mmp30a*)
AS13	ACTGCGACAAATACGCCTCT	

## Data Availability

The original contributions presented in this study are included in the article. The raw RNA-Seq data were deposited into the NCBI data bank with the accession number PRJNA1082072. Further inquiries can be directed to the corresponding authors.

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
