# Peer review of "Ace Deficiency Induces Intestinal Inflammation in Zebrafish"

_ijms, 2024, doi:10.3390/ijms25115598_

Round 1
Reviewer 1 Report
Comments and Suggestions for Authors
This research uncovers the pivotal role of angiotensin-converting enzyme (ACE) in maintaining intestinal homeostasis, shedding light on its significance in the pathogenesis of Inflammatory Bowel Disease (IBD). Through zebrafish models, the study demonstrates that ACE deletion leads to intestinal inflammation and increased susceptibility to enteritis, highlighting ACE as a potential therapeutic target for IBD management.
Manuscript can be accepted once the minor issues are corrected.
1. In abstract Keywords:- Provide catchy words which are not used in the title of the research.
2. Line 56-60:- Refine the research aim by articulating detailed objectives in a point-by-point format.
3. Line 67:- “D. rerio” should be in italics also check the basic typo- and spacing errors throughout the manuscript.
4. Line 77: - Expand NCBI for the first time.
5. Line 88-89:- Please specify the country of origin for the purchased Stereomicroscope.
6. Line 103: - provide the information about the Image J software in the brackets.
7. Please ensure that a scale bar is included in all histogram images.
8. Figure 7A: - Enhance the clarity of the images for WT and ace-/- to improve their visual appeal, as they appear slightly dull.
9. Line 375-380: Section 5:- The study's conclusion should summarize key findings and potential future directions.
Comments on the Quality of English LanguageThe grammar is fine, some typo- and spacing errors needs to be corrected.
Reviewer 2 Report
Comments and Suggestions for Authors
Review for: Ace Deficiency Induces Intestinal Inflammation in Zebrafish
Abstract and introduction
· Line 50 – The authors state that they are referencing “recent” research on ACE, but the papers cited are from 2005 and 2008. Was this paper shelved for a time? No issue if it was I would just ensure the literature is up to current.
Methods and results
· I see not issue with the methodology or the results. The study design and analysis was appropriate to address the role of ACE in intestinal inflammation and is appropriately performed to the best of my knowledge.
· Graphs and images are of high quality and good resolution and display the data very well.
Discussion and conclusion
· These sections and the following sections for data availability, author contributions, ect. have no apparent issues.
I recommend publication of this manuscript. It is well done and fits the scope of the journal.
